# Vibrational control of selective bond cleavage in dissociative chemisorption of methanol on Cu(111)

Jialu Chen [1], Xueyao Zhou [1], Yaolong Zhang [1] & Bin Jiang [1]

Controlling product branching ratios in a chemical reaction represents a desired but difficult achievement in chemistry. In this work, we demonstrate the first example of altering the branching ratios in a multichannel reaction, i.e., methanol dissociative chemisorption on Cu (111), via selectively exciting specific vibrational modes. To this end, we develop a globally accurate full-dimensional potential energy surface for the $CH_3OH/Cu(111)$ system and perform extensive vibrational state-selected molecular dynamics simulations. Our results show that O–H/C–H/C–O stretching vibrational excitations substantially enhance the respective bond scission processes, representing extraordinary bond selectivity. At a given total energy, the branching ratio of C–O/C–H dissociation can increase by as large as 100 times by exciting the C–O stretching mode which possesses an unprecedentedly strong vibrational efficacy on reactivity. This vibrational control can be realized by the well-designed experiment using a linearly polarized laser.

[1] Hefei National Laboratory for Physical Science at the Microscale, Department of Chemical Physics, University of Science and Technology of China, Hefei, Anhui 230026, China. Correspondence and requests for materials should be addressed to B.J. (email: bjiangch@ustc.edu.cn)

Control of product branching ratios in a chemical reaction is one of the ultimate goals in chemistry. Selectively exciting a specific reagent vibrational mode is one possible way toward this goal, because of different effects of vibrational excitations on the reactivity relative to the translational excitation[1]. Thanks to advances in laser technology, this so-called mode specificity and the related bond selectivity have been first explored in gas-phase reactions[2,3], and are now believed to be general in gas–surface reactions as well[4,5]. For example, as rate-limiting steps in methane steaming reforming and water–gas shift processes, respectively, methane and water dissociative chemisorption has been extensively investigated by state-of-the-art quantum-state-resolved molecular beam experiments[4,6–9] and by first-principles quantum/classical dynamical calculations[10–16]. It is well established that the dissociative sticking probabilities of $CH_4$ and $H_2O$ on a variety of single-crystal metal surfaces exquisitely depend on specific vibrational excitations. Very recently, the dissociative chemisorption of $CO_2$ on Ni(100) has also been predicted to be mode specific[17].

These studies have provided unprecedented details of the nonstatistical nature of gas–surface reactions, highlighting the importance of reaction dynamics. It has been argued that the intramolecular vibrational energy redistribution (IVR) is far from complete prior to molecular dissociation at the surface[4]. As a result, mode-specific reactivity can be reasonably rationalized by a transition state-based model in the sudden limit that relies on the coupling between a vibrational motion and the reaction coordinate[14,18]. Alternatively, mode specificity may also be ascribed to the mode softening and vibrational nonadiabatic couplings in the reaction path Hamiltonian (RPH) wavepacket model developed by Jackson and coworkers[11]. In spite of significant progress, our understanding of the mode-specific chemistry is yet far from complete.

To date, detailed mode-specific reaction dynamics have been mostly demonstrated in molecules containing only a single type of chemical bond (or isotopically substituted) to be broken (i.e., O–H bond in $H_2O$ or C–H bond in $CH_4$). For true bond selectivity, the cleaved bonds are different. It is still unclear that how a specific vibrational excitation would selectively manipulate the branching ratio of multiple bond-breaking processes in a polyatomic molecule at the surface. To answer this question, the underlying multidimensional potential energy surface (PES) globally covering multiple reaction channels is needed, but it is very challenging for gas–surface systems.

Methanol is a promising next generation of energy carrier that can be used for on-board hydrogen production and/or in direct methanol fuel cells, thanks to its easy storage and transportation requirements and high H/C ratio. The dissociative chemisorption of methanol on metal surfaces is the initial and key step for hydrogen production from methanol[19]. For our purposes, more importantly, methanol dissociation on metal surfaces is an ideal model system for better understanding the mode-specific and bond-selective chemistry since it involves the cleavage of three classes of chemical bonds (C–H, C–O, and O–H). It also represents a prototype of reactions of complex organic molecules on metal surfaces[20].

It is generally accepted that metallic copper is the active component in the commonly used Cu-based catalysts dispersed on oxide support for methanol synthesis, decomposition, as well as steam-reforming processes[19]. As a consequence, much concern has been concentrated on reaction mechanisms involving methanol on copper[21–35]. Earlier experimental studies revealed that methanol adsorbs molecularly on most low-index copper surfaces at low temperatures, while pre-covered oxygen atoms were found to significantly facilitate methanol decomposition[21–23]. Periodic density functional theory (DFT) calculations provided consistent

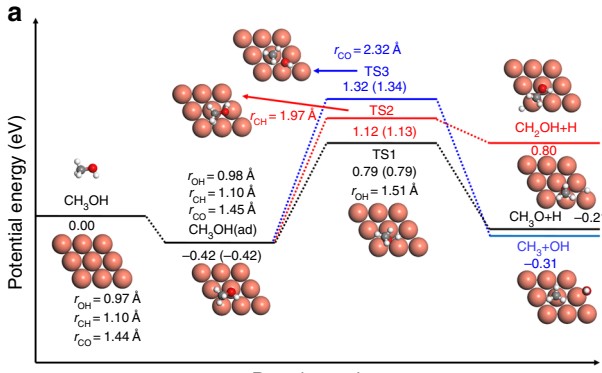

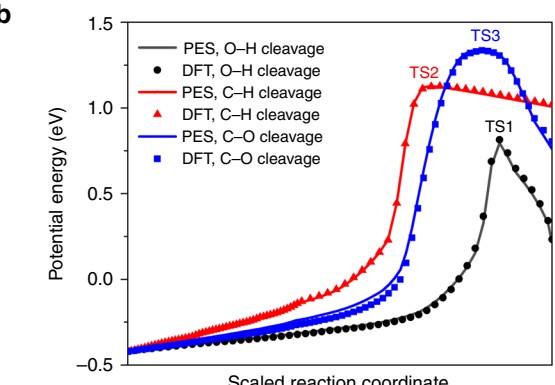

**Fig. 1** Reaction profile. **a** Energetics (in eV) and stationary point geometries (in Å) of O–H (black), C–H (red), and C–O (blue) bond cleavages for $CH_3OH$ dissociative chemisorption on Cu(111) computed by DFT. Energies in parentheses are obtained from the PES for comparison. **b** Minimum energy paths for these three dissociation channels on the permutation invariant polynomial neural network potential energy surface (solid lines) and DFT energies (symbols) recalculated at selected points along the respective reaction coordinate. Note that the reaction coordinate has been scaled to be in the same range

information that methanol dissociation is highly activated on Cu (111) and Cu(110)[29,30,34,35]. The O–H dissociation channel is found to be most favorable, while the C–H and C–O bond scissions are very difficult. According to Polanyi's rules[36], these remarkable and product-like ("late") dissociation barriers are likely to manifest mode specificity, as already demonstrated in the activated adsorption of $CH_4$ and $H_2O$ on metal surfaces[14]. However, very few studies have focused on the dissociation dynamics of methanol[37,38]. The only molecular beam experiments indicated that methanol decomposition on Pt(111)[37] and Ru/V alloy[38] occurs at low-incidence energies with the O–H bond cleavage via a molecular precursor mechanism, in which the IVR may take place because of the long residence time and the mode specificity is less likely. To explore the potential ability of vibrational control of reactivity of a chemical reaction, we theoretically investigate the mode-specific and bond-selective reaction dynamics in $CH_3OH$ dissociative chemisorption on Cu(111).

## Results

**Reaction pathways and potential energy surface.** In Fig. 1a, three possible reaction pathways for methanol dissociative chemisorption on the rigid Cu(111) are compared. Prior to dissociation, methanol is found to molecularly adsorb on the surface preferentially on the top site through the oxygen atom, consistent

with previous theoretical results[26,29–35]. The calculated Cu–O separation is 2.39 Å and the corresponding binding energy is −0.42 eV, indicating a weak binding character, in good agreement with the experimental values (2.69 Å[39] and −0.48 eV[40]). The methanol-surface interaction predicted here is stronger than previous DFT predictions[33,36–42], suggesting the importance of including the van der Waals effects.

The dissociation of methanol on Cu(111) is highly activated. The O–H bond scission leads to methoxy and hydrogen atoms via the smallest barrier (labeled as TS1) of 0.79 eV with respect to free CH₃OH plus Cu(111), while the barriers for breaking C–H (TS2) and C–O (TS3) bonds are 0.33 and 0.53 eV higher, respectively. Interestingly, all the transition states feature the "late" barrier characters, with the dissociating O–H, C–H, and C–O bonds substantially extended to 1.51, 1.97, and 2.32 Å, respectively. The elongation of dissociative bonds at transition states has been observed for similar $H_2O$[9], $CH_4$[10], and $CO_2$ dissociations[17] on various metal surfaces, which is responsible for the vibrational enhancement on reactivity as suggested by Polanyi's rules.

It can be seen in Fig. 1a and in additional results of Supplementary Figs 1 and 2, Supplementary Tables 1–3 and Supplementary Discussion that the permutation invariant polynomial neural network (PIP-NN) PES accurately reproduces these energetics and geometries, as well as frequencies of the stationary points. The quality of the PIP-NN PES is further supported by the excellent agreement between the predicted energies on the PES and those recalculated by DFT, at identical selected geometries on minimum energy paths (MEPs), as displayed in Fig. 1b. Interestingly, the intrinsic reaction coordinate search starting from any transition state finds ultimately the same adsorption well on the reactant side.

Figure 2 demonstrates the two-dimensional contour plots of the PES as a function of the dissociative and translational coordinates for all of three dissociation channels, with other internal coordinates involved in the reaction and molecular lateral position optimized. It is manifest that the PES well represents the dynamically relevant configuration space without unphysical holes. Both the adsorption well and "late" transition states are clearly displayed, which again validates the accuracy of the PIP-NN fit. Following the MEPs, the CH₃OH molecule would first adsorb weakly in the pre-reaction well and then select one of the competitive dissociation pathways. It is emphasized that the differences of the potential topography for different reaction channels may affect the translational-to-vibrational energy transfer[41], as discussed below.

**Dissociation probabilities**. The calculated probabilities for breaking the O–H, C–H, and C–O bonds in the ground state of CH₃OH as a function of incidence energy ($E_i$) are compared in Fig. 3. With the increasing incidence energy, reaction probabilities increase monotonously, reflecting the directly activated nature of all pathways. Given the trend of barrier height, namely $E_b$ (O–H) < $E_b$ (C–H) < $E_b$ (C–O), it is readily understood that O–H dissociation is the most favorable, followed by the C–H and C–O dissociations orderly. Interestingly, the O–H bond starts to break (e.g., $P_0 = 10^{-3}$) at $E_i \approx 0.75$ eV, close to its barrier height. However, the translational energy thresholds for the other two dissociation channels (1.5 eV for C–H and 2.5 eV for C–O) are much higher than their respective barriers. Especially, the C–O bond is extremely inert and its dissociation probability is very low, and even the incidence energy is nearly twice as the barrier height. Our results indicate that the C–H and C–O dissociation channels not only have higher barriers but also require more translational energy above the barrier to open. As a result, the corresponding products are extremely unfavorable in thermal

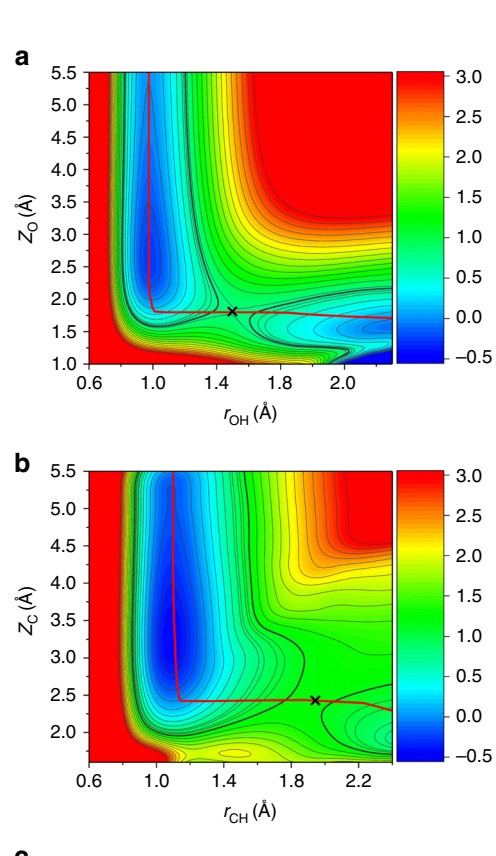

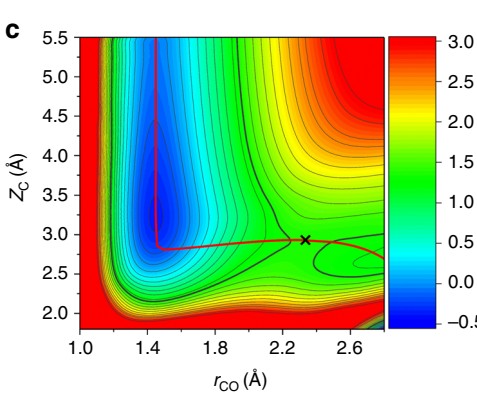

**Fig. 2** Two-dimensional potential energy contour plots. O–H (**a**), C–H (**b**), and C–O (**c**) dissociation channels are illustrated as a function of $Z_O$ and $r_{OH}$ (**a**), $Z_C$ and $r_{CH}$ (**b**), and $Z_C$ and $r_{CO}$ (**c**), respectively, with other coordinates optimized. Red solid lines and black crosses correspond to the projected MEPs along reaction coordinates and transition states, respectively. The colored energy column is given in eV

conditions, in line with experimental findings that only CH₃O was observed on Cu(110)[22,23]. The ineffectiveness of channeling translational energy into a reaction coordinate has hitherto been observed in the site-specific dissociation of $H_2O$ on Ni(111)[41] and HCl on Au(111)[42], which both originate from the topography of the PES. From Fig. 2a–c, corresponding to O–H, C–H, and C–O bond scissions, respectively, the energy flow from the translational to each dissociation coordinate becomes increasingly difficult because of the shrink of the angle characterizing the "elbow" PESs. Furthermore, bond lengths of the dissociating O–H, C–H, and C–O at transition states are incrementally elongated. Both features promote the possibility of recoil when a molecule impacts on the surface with high translational velocity, thus resulting in the increasing threshold energy to open these three channels.

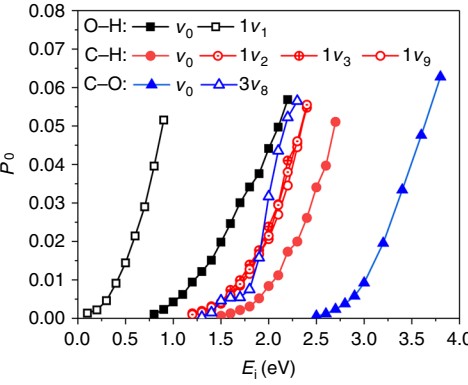

**Fig. 3** Dissociation probabilities. Calculated O–H/C–H/C–O dissociation probabilities as a function of normal incidence energy of CH$_3$OH($v$) on Cu (111) in its ground ($v_0$) and first excited states with O–H stretching ($1v_1$), C–H symmetric stretching ($1v_3$), C–H antisymmetric stretching ($1v_2$ (A') and $1v_9$ (A")), and three quanta overtone with C–O stretching ($3v_8$), respectively

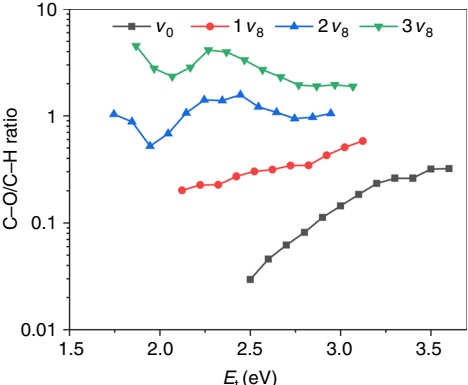

**Fig. 4** Branching ratios. C–O/C–H bond dissociation ratio as a function of total energy with respect to the potential energy of free CH$_3$OH molecule plus its zero-point energy, varying with the incremental excitation of the C–O stretching mode ($v_8$)

**Table 1 Calculated vibrational efficacies with respect to the overall reactivity of these influential vibrational modes in the O–H/C–H/C–O dissociation**

| Dissociation probabilities of (O–H/C–H/C–O) | 0.001 | 0.005 | 0.01 | 0.02 | 0.05 |
|---|---|---|---|---|---|
| $1v_1$(O–H) | 1.62 | 1.64 | 1.72 | 1.97 | 2.66 |
| $1v_2$(C–H) | 0.95 | 0.86 | 0.89 | 0.92 | 0.94 |
| $1v_3$(C–H) | 1.10 | 1.00 | 1.01 | 1.01 | 1.01 |
| $1v_9$(C–H) | 1.00 | 0.80 | 0.84 | 0.87 | 0.94 |
| $1v_8$(C–O) | 5.13 | 5.00 | 4.21 | 4.15 | 5.25 |
| $2v_8$(C–O) | 4.45 | 4.33 | 4.31 | 4.26 | 4.12 |
| $3v_8$(C–O) | 3.57 | 3.37 | 3.45 | 3.76 | 4.31 |

**Effects of reactant vibrations on each reaction channel**. On the other hand, vibrational excitations can dramatically change this picture since different vibrational states have different impacts on the reaction. The vibrational enhancement of the reactivity over translational enhancement can be quantified by the vibrational efficacy, which is often defined by $\eta(v) = [E_i(0, P_0) - E_i(v, P_0)]/\Delta E_v$,[43] where $E_i(0, P_0)$ and $E_i(v, P_0)$ are translational energies for a given reaction probability $P_0$ of the ground (0) and vibrationally excited ($v$) states, and $\Delta E_v$ the corresponding vibrational energy. A vibrational efficacy larger than unity illustrates that vibration is more effective in enhancing the reaction than translation. In this work, we concentrate on the influence of stretching mode excitations and compare the vibrational efficacies of the modes with similar vibrational energies, i.e., the first excited state in O–H stretching ($1v_1 = 3670$ cm$^{-1}$), C–H symmetric ($1v_3 = 2887$ cm$^{-1}$) and asymmetric stretching ($1v_2$ (A') $= 2979$ cm$^{-1}$ and $1v_9$ (A") $= 2935$ cm$^{-1}$) modes, and three quanta overtone with C–O stretching ($3v_8 = 2970$ cm$^{-1}$). It should be noted that, unlike methane, none of the vibrational states in methanol is degenerate.

In Fig. 3 and Supplementary Figs 3–5, it is found that exciting a specific bond stretching mode drastically promotes the corresponding bond cleavage, while it hardly affects the other bond-breaking processes, thus representing a strong bond selectivity. For example, as listed in Table 1, $1v_1$ state shows a considerable enhancement for O–H dissociation with $\eta(1v_1, O-H)$ being in the range from 1.62 to 2.66 depending on translational energy

considered in this study. This is a fairly large value compared to that of methane and water dissociation[14]. However, $1v_1$ state has a very minor effect on C–H and C–O scissions, for which the vibrational efficacies are close to zero (see Supplementary Table 4). In addition, excitations in the symmetric ($1v_3$) and asymmetric ($1v_2$ and $1v_9$) C–H stretching modes are more or less as effective as translation in promoting the C–H dissociation, with the former having a slightly higher vibrational efficacy because of its lower frequency. This is similar to the observations for methane dissociation[4].

The most pronounced vibrational efficacy is found for the C–O stretching mode which almost exclusively facilitates the C–O dissociation. As seen in Supplementary Table 4 and Supplementary Fig. 5, the single quantum excitation of $v_8$ leads to an incredible $\eta(1v_8, C-O)$ ranging from 4.15 to 5.21. To the best of our knowledge, this is so far the largest vibrational efficacy known among various molecular dissociations on solid surfaces. In comparison, while vibrational efficacies of both symmetric and antisymmetric C–O stretching modes in CO$_2$ dissociation on Ni (100) are already quite large, i.e., 1.8~2.2, they are still much smaller than $\eta(1v_8, C-O)$ observed here. Although $\eta(v)$ typically decreases with multi-quanta excitations as seen in methane and water dissociations[4,9], $\eta(2v_8, C-O)$ and $\eta(3v_8, C-O)$ here are still as large as 4.12~4.45 and 3.37~4.31, respectively. As a result, the dissociation probability of C–O in the $3v_8$ state becomes comparable to that of C–H in the $1v_3$, $1v_2$, or $1v_9$ state, given the very close total energy in these states.

As more clearly seen in Fig. 4, such tremendous vibrational efficacies for the first three excited states of $v_8$ mode gradually invert the branching ratio for C–H and C–O dissociation channels at a fixed total energy ($E_t$, the sum of translational and vibrational energy with respect to the zero-point energy of CH$_3$OH). At the total energy of 2.5 eV, for example, the C–H/C–O branching ratio increases roughly ten times from ~0.03 to ~0.30 via $1v_8$ excitation. The two-quanta $v_8$ excitation makes C–H and C–O dissociation processes nearly equally weighted, and $3v_8$ excitation further promotes the C–H/C–O ratio making the C–O channel dominant, resulting in an overall ~100 times enhancement compared to the ground state value. This is an encouraging example that demonstrates the possibility of altering the branching ratios via vibrational excitations in a practical multi-channel reaction, which awaits experimental validation. We note in passing that the lattice motion and electron–hole pair excitations neglected in this work may change the dissociation probability to a lesser extent in the energy range considered here,

**Table 2 Sudden vector projection values of stretching vibrational modes and translation along surface normal associated with three transition states for various bond scissions**

| Symmetry | Mode | Labels | Sudden vector projection values | | |
|---|---|---|---|---|---|
| | | | TS1(O–H) | TS2(C–H) | TS3(C–O) |
| A′ | OH stretching | $\nu_1$ | 0.869 | 0.033 | 0.013 |
| A′ | CH₃ as-stretching | $\nu_2$ | 0.003 | 0.178 | 0.023 |
| A″ | CH₃ as-stretching | $\nu_9$ | 0.003 | 0.515 | 0.087 |
| A′ | CH₃ s-stretching | $\nu_3$ | 0.005 | 0.508 | 0.104 |
| A′ | CO stretching | $\nu_8$ | 0.026 | 0.015 | 0.745 |
| | Translation $z$ | | 0.121 | 0.100 | 0.242 |

but previous results have shown that they do not qualitatively alter the mode specificity obtained in the static surface calculations[44–46].

## Discussion

Mode specificity and bond selectivity observed in CH₃OH dissociation on Cu(111) can be readily understood by the so-called sudden vector projection (SVP) model[47,48]. As detailed in the Supplementary Methods, this SVP model relates the effect of reactivity of a reactant mode to the overlap between the reactant mode vector ($\mathbf{Q}_i$) and the reaction coordinate vector at the transition state ($\mathbf{Q}_{RC}$), namely, $p_i = \mathbf{Q}_i \cdot \mathbf{Q}_{RC} \in [0, 1]$. We computed here an individual set of SVP values for each reaction channel separated by respective transition states. Part of them are listed in Table 2 and a more complete list can be found in Supplementary Table 5.

First, it is worth noting that O–H, C–H, and C–O stretching modes are exclusively coupled with the corresponding reaction coordinates. For example, the $\nu_1$ mode has a particularly largest SVP value (~0.87) with TS1 and other vibrational and translational modes all have small SVP values. Similarly, for C–O dissociation, the $\nu_8$ mode couples with the reaction coordinate of the C–O dissociation barrier most strongly with $p_8 = 0.75$, which is much larger than the SVP values of the translational modes. These results agree well with remarkable vibrational efficacies of the respective local bond-stretching excitations. On the other hand, the couplings between normal modes and the reaction coordinate for C–H dissociation are generally weaker, consistent with the smaller vibrational efficacies. Specifically, the C–H symmetric ($\nu_3$) and one of the two asymmetric stretching $\nu_2$ (A′) modes possess similar SVP value of ~0.51, while another asymmetric stretching $\nu_9$ (A″) mode couples with the reaction coordinate more weakly with $p_9 = 0.18$. However, our QCT results show analogous enhancements of $\nu_2$ (A′) and $\nu_9$ (A″) modes that are close in energy but in different symmetries. A plausible explanation is that two states mix with each other because of their near-degenerate frequency in our quasi-classical treatment, as observed for those threefold degenerate modes in CH₄[49,50]. Experimentally, Beck and coworkers have recently observed the important role of symmetry in the nearly isoenergetic vibrations in dissociative chemisorption of CH₄($2\nu_3$) on Pt(111)[51]. As discussed in refs. [12,52] on CHD₃ dissociation, the reliability of QCT results decreases when there exist strong vibrational state couplings and energy flow in the entrance channel. An accurate interpretation of such a phenomenon requires a full-dimensional quantum mechanical description of symmetry, which is however currently too expensive if not impossible and beyond the scope of this article.

In conclusion, we have developed the first eighteen-dimensional PES for the methanol dissociative chemisorption on a Cu(111) surface with an analytical PIP-NN representation fitted with over 200 thousand DFT points. This globally accurate PES not only includes all molecular degrees of freedom but also describes the multi-channels for O–H, C–H, and C–O dissociations equivalently well. The distinct topography along each reaction pathway determines the effectiveness of translating the incident energy into the reaction coordinate and results in the very different translational energy threshold of each channel. QCT calculations on this PES indicate that the C–O bond breaking is neither kinetically (high barrier) nor dynamically (high threshold energy) favorable, having a much lower probability than the other two channels in the ground state CH₃OH. It is also revealed that the excitation of the O–H/C–H/C–O stretching mode significantly facilitates the corresponding bond dissociation process, representing an unambiguous mode specificity and bond selectivity. Because of its unprecedented vibrational efficacy for C–O scission, multi-quanta C–O stretching excitations are able to invert the C–O/C–H branching ratio. The C–O and C–H dissociation probabilities become comparable at a given total energy with C–H ($1\nu_3/1\nu_2/1\nu_9$) and C–O ($3\nu_8$) stretching modes excited. To the best of our knowledge, this is the first demonstration of the vibrational control of the branching ratios of a reaction with multiple channels. It is expected to greatly enrich our understanding to mode-specific chemistry and offer a potential means to manipulate product branching ratios in chemical reactions. We hope that these detailed predictions can motivate further experimental tests.

## Methods

**Electronic structure calculations**. Taking all molecular degrees of freedom into account, we report here the first globally accurate eighteen-dimensional (18D) PES for methanol dissociation on Cu(111), in which O–H, C–H, and C–O bond scissions are equally well described. Plane wave density functional theory (DFT) calculations were performed via the Vienna Ab initio Simulation Package (VASP)[53]. The Cu(111) surface was represented by a slab model with 3 × 3 unit cells and four layers, in which the top two layers were optimized and then fixed at equilibrium positions. The ion–electron interactions were described using the projector-augmented wave (PAW) method[54]. The plane wave basis was truncated at kinetic energy of 400 eV. The electron exchange correlation was described by the optPBE-vdW functional[55], including dispersion correction, which was necessary to reproduce the experimental binding energy. The 5 × 5 × 1 Monkhorst–Pack $k$-point grid was tested to converge the binding and activation energies within 0.05 eV.

**Potential energy surface**. To develop the 18D PES incorporating the surface periodicity and permutation symmetry in the molecule, the recently proposed permutation-invariant polynomial-neural network (PIP-NN) approach[56,57] was employed. Specifically, 33 primitive functions satisfying the periodicity were initiated to yield 546 PIPs up to the sixth degree using the SINGULAR software[58], serving as the input of NNs. The CH₃OH/Cu(111) configurations were sampled iteratively to cover all three dissociation channels and converge the dissociation probabilities. Over 200,000 points were finally collected and fitted to the two-layer NNs with 18 and 100 neurons in the first and second hidden layers, respectively, based on a hybrid extreme learning machine Levenberg–Marquardt algorithm[59]. The resultant PES was an average of the three best fits yielding a root-mean-square error (RMSE) of 37.2 meV. More details of the PES fitting, data sampling, and convergence tests can be found in the Supplementary Methods.

**Quasi-classical trajectory calculations**. Since a fully coupled quantum dynamical treatment of this system is still a forbidden task, we computed initial state-selected

dissociation probabilities ($P_0$) employing the quasi-classical trajectory (QCT) method. Electron–hole pair excitations have been found to have a minor effect in directly activated reactions[44,60], and have been thus neglected here. Although a more quantitative comparison to an experiment may require a quantum mechanical treatment with inclusion of lattice motion[45], the QCT approach, which has proven to capture the essence of mode specificity at collision energies above the barrier heights and well reproduce the bond selectivity when exciting a local stretching mode[61], is adequate for our present purposes here. In addition, recent QCT-based ab initio molecular dynamics applications have reproduced measured dissociative sticking coefficients well above reaction barriers for CHD$_3$ on both flat (Ni(111) and Pt(111)) and stepped (Pt(211)) surfaces within chemical accuracy[12,52]. Overall, up to $10^6$ trajectories have been run in a wide range of translational energies for different initial states. More details and additional results are given in the Supplementary Methods.

## Data availability

The data that support the findings of this study are available from the corresponding author upon request.

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

## Acknowledgements

This work was supported by National Key R&D Program of China (2017YFA0303500), National Natural Science Foundation of China (21573203, 91645202, and 21722306), and Anhui Initiative in Quantum Information Technologies. We appreciate the Super-computing Center of USTC and AM-HPC for high-performance computing services. We thank Prof. Hua Guo for many helpful suggestions.

## Author contributions

B.J. designed this project, J.C. performed the first-principles calculations and PES fitting, and X.Z. and Y.Z. helped on the data sampling and NN codes. J.C. and B.J. wrote the manuscript. All authors read and commented on the manuscript.

## Additional information

**Competing interests:** The authors declare no competing interests.

