## [Peer Review File · Nature Communications]

Reviewers' comments:

Reviewer #1 (Remarks to the Author):

The manuscript focuses on a model gas-surface reaction, i.e. the dissociative chemisorption of methanol on a copper surface. The authors explore the reactivity of the system by means of quasi-classical trajectory (QCT) simulations based on a full-dimensional potential energy surface (PES), which has been fitted to thousands of DFT energy points. In particular, state-selected reaction probabilities are calculated for various pre-excited vibrational states. Results suggest mode-specificity with particularly high vibrational efficacies and bond-selectivity with branching ratios far from statistical.

The manuscript is overall well written and the conclusions are well supported by data. However, I am afraid I cannot recommend publication in a high-impact journal such as Nature Communications, as the work presented lacks of sufficient novelty. First of all, the main focus of the paper is on a feature of molecule-surface reactions that has already been reported and thoroughly investigated for similar systems: the role played by vibrational pre-excitation on the product distribution. Similar bond-selective behaviors, in fact, have already been explored for both methane (ref. 8 of the main paper and ref [1] below) and water (e.g. refs. [2] and [3] below), which limits the impact of similar findings for methanol. The methodology is also not novel but well-established: the authors themselves have already published numerous QCT studies involving neural-network-based PESs for both gas-phase and gas-surface reactions (e.g. refs. 10-16 of the main paper). The number of degrees of freedom included in the PES (18) might be the largest considered so far in a molecule-surface reaction, but this does not represent a significant advance with respect, for instance, to the 15-dimensional PES for CH₄ that has been constructed by the authors following the same strategy (ref. 58 of the main paper). Finally, the manuscript does not provide significant new insight in the methanol dissociation process: the predicted reaction probabilities are likely not quantitatively accurate, as also stated by the authors on line 221, and the absence of experimental data for the system does not allow for the validation of the various approximations that are part of the dynamical model (static-surface approximation, electronic adiabaticity, classical treatment of nuclear motion). The proposed analysis of the molecule's dissociation dynamics also represents a rather 'routine' task in the field of gas-surface reactions (see again refs. 10-16 for similar examples).

For all these reasons, I believe that the manuscript does not report 'important advances of significance to specialists', as stated in the scope of the journal. The manuscript should be suitable instead for a more technical journal.

[1] Chen et al., Faraday Discuss. 157, 285 (2012).

[2] Jiang et al., Chem. Sci. 4, 503 (2013).

[3] Farjamnia and Jackson, JCP 142, 234705 (2015).

Reviewer #2 (Remarks to the Author):

The authors have constructed a global full (18) degree-of-freedom potential energy surface for the reaction of CH₃OH on a rigid Cu(111) surface. This is done using a neural network approach, fitting to over 200,000 DFT energies. These energies are computed using a 3x3 4-layer supercell and a 5x5x1 grid of k-points, which is very reasonable and much more accurate than is typically done for these sorts of big calculations. QCT methods are used to compute the probabilities for C-H, C-O and O-H bond scission as a function of collision energy for a variety of initial vibrational states.

Overall, this is an impressive calculation and the results are very interesting. I recommend

publication, for several reasons, after the authors have considered my minor comment below. The paper is well written and should be of interest to chemists in a variety of fields. This is the most complicated gas-surface reaction studied in this manner to date, and a big step beyond CH₄ reactions, which have been intensively studied for almost two decades now. The dissociation of methanol on a metal-based catalyst is an important step in several important processes, and this is the first theoretical study of the dynamics of this reaction. In addition, there has been much interest in bond-selective chemistry, but until now, theory and experiment have focused only on isotopologues of water and methane, such as HOD and CHD₃. That is, they have examined X-H vs X-D bond cleavage. This is the first study, theoretical or experimental, to go beyond this, to my knowledge, with three different bonds breaking. And finally, the vibrational efficacies for promoting dissociation of the various bonds are unusually large. I'm not aware of efficacies this large having been either computed or measured for any system. In addition, the A' and A'' C-H stretching modes have a similar efficacy, which is very unusual. I would have expected the A'' mode to have a very small efficacy. This paper will certainly stimulate much work in the field.

1. At two points in the manuscript the authors note that lattice motion effects are likely to modify the dissociation probabilities, listing reference 16 or 14. I think it fair to reference either the original Tiwari/Jackson paper(s), or perhaps the more recent 2016 Guo/Afarjamnia/Jackson Perspective in JPC Letters discussing this topic.

Reviewer #3 (Remarks to the Author):

The authors describe a computational study of methanol decomposition dynamics on a Cu(111) surface. They first perform a density functional theory-based electronic structure calculation and fit the resulting grid of calculated points to obtain a high-dimensional potential energy surface for describing the reaction. Reaction dynamics on this potential point to very late barriers relative to the C-H, O-H, and C-O stretching vibrations. Since reactant distortion along each of these coordinates leads to chemically distinct reaction products, the authors conclude that there is a strong likelihood that vibrational state selected beam-surface scattering experiments will uncover unusually strong mode- and bond-selective chemistry. The work is novel in that it advances computational state-of-the-art in molecule-surface scattering calculations to a molecule that is larger and more structurally complex than any previously studied. Ambitious studies such as this are the drivers that will move the field forward. The work provides a set of detailed predictions that experiment can test. More broadly, the work is at the forefront of attempts to develop increasingly accurate computational methods for predicting barrier heights and reaction paths for important heterogeneously catalyzed processes, and for using those calculations to screen for and predict new catalysts with optimized selectivity and activity. The work also represents an important advance in the ability of computational chemistry to extend chemically accurate predictions of reactivity to larger, and more complex chemical systems.

To enable this calculation, the authors relied on a number of established computational methods to obtain the potential energy of the system in 18D at a number of fixed points, and a recently reported neural-network method to fit a potential to these points. Quasi-classical trajectories are then calculated to gain insight into reaction dynamics. Citations to the relevant precedents are included.

To me, two aspects of the paper really stand out. First, the authors demonstrate that with currently available computational hardware and algorithms, highly detailed and high-dimensional studies of gas-surface reaction dynamics have become accessible to study. Second, I felt that the authors did an excellent job of surveying prior work in the field, and of designing and presenting their work in a way that maximizes its impact on the field.

The authors are to be commended for the extensive scope of their calculation. A strong propensity for mode- and bond-selective chemistry could have been predicted from a much more

straightforward transition state calculation and application of Polanyi's rules. Manifestation of that propensity into non-statistical behavior depends on how closely the gas-phase molecule's vibrational eigenstate, which may result from significant state-mixing among zero-order vibrational normal modes, resembles the vibrational state used in the calculation. Non-statistical behavior also depends on the extent of vibrational state mixing that occurs in the entrance channel for the reaction - behavior that arises from, and must be modeled by quantum effects.

Therefore, the quantitative accuracy of the work is likely limited by the same factors associated with prior studies using this general approach. While QCT-based calculations have proven useful for predicting reaction probabilities for trajectories well above reaction barriers, they fail to predict, with quantitative accuracy, reactivity for system energies most important to thermal processes - i.e. trajectories very near in energy to the reaction barrier. Details of vibrational state coupling and energy flow in the entrance channel, which depend strongly on quantum effects missing in the QCT approach, and have been shown to result in decreased accuracy of the computed results. Two recent papers discuss this point, and should be included in the authors' discussion of their results. (D. Migliorini, H. Chadwick, F. Nattino, A. Gutiérrez-González, E. Dombrowski, E. A. High, H. Guo, A. L. Utz, B. Jackson, R. D. Beck, and G.-J. Kroes, *Surface Reaction Barriometry: Methane Dissociation on Flat and Stepped Transition Metal Surfaces* *J. Phys. Chem. Lett.* 8, 4177 (2017) and F. Nattino, D. Migliorini, G.-J. Kroes, E. Dombrowski, E. A. High, D. R. Killelea, and A. L. Utz, *Chemically accurate simulation of a polyatomic molecule-metal surface reaction* *J. Phys. Chem. Lett.* 7, 2402 (2016))

In summary, the authors describe an ambitious computational study based on the implementation of established electronic structure and QCT methods. While its significance would have been bolstered by associated experimental work, its detailed predictions and conclusions are exactly what is needed to motivate experimental tests and advance the field.

Reviewers' comments:

Reviewer #1 (Remarks to the Author):

The manuscript focuses on a model gas-surface reaction, i.e. the dissociative chemisorption of methanol on a copper surface. The authors explore the reactivity of the system by means of quasi-classical trajectory (QCT) simulations based on a full-dimensional potential energy surface (PES), which has been fitted to thousands of DFT energy points. In particular, state-selected reaction probabilities are calculated for various pre-excited vibrational states. Results suggest mode-specificity with particularly high vibrational efficacies and bond-selectivity with branching ratios far from statistical.

The manuscript is overall well written and the conclusions are well supported by data. However, I am afraid I cannot recommend publication in a high-impact journal such as Nature Communications, as the work presented lacks of sufficient novelty. First of all, the main focus of the paper is on a feature of molecule-surface reactions that has already been reported and thoroughly investigated for similar systems: the role played by vibrational pre-excitation on the product distribution. Similar bond-selective behaviors, in fact, have already been explored for both methane (ref. 8 of the main paper and ref [1] below) and water (e.g. refs. [2] and [3] below), which limits the impact of similar findings for methanol. The methodology is also not novel but well-established: the authors themselves have already published numerous QCT studies involving neural-network-based PESs for both gas-phase and gas-surface reactions (e.g. refs. 10-16 of the main paper). The number of degrees of freedom included in the PES (18) might be the largest considered so far in a molecule-surface reaction, but this does not represent a significant advance with respect, for instance, to the 15-dimensional PES for CH₄ that has been constructed by the authors following the same strategy (ref. 58 of the main paper). Finally, the manuscript does not provide significant new insight in the methanol dissociation process: the predicted reaction probabilities are likely not quantitatively accurate, as also stated by the authors on line 221, and the absence of experimental data for the system does not allow for the validation of the various approximations that are part of the dynamical model (static-surface approximation, electronic adiabaticity, classical treatment of nuclear motion). The proposed analysis of the molecule's dissociation dynamics also represents a rather 'routine' task in the field of gas-surface reactions (see again refs. 10-16 for similar examples).

For all these reasons, I believe that the manuscript does not report 'important advances of significance to specialists', as stated in the scope of the journal. The manuscript should be suitable instead for a more technical journal.

[1] Chen et al., Faraday Discuss. 157, 285 (2012).

[2] Jiang et al., Chem. Sci. 4, 503 (2013).

[3] Farjamnia and Jackson, JCP 142, 234705 (2015).

Author reply: We respectfully disagree this referee's assessments. It is true that our model is not perfect and contains some approximations, however, we would like to emphasize that our work does present a few significant advances in this field.

First of all, the increase of molecular size is actually not the most notable difference between our work and previous studies on the dissociative chemisorption of methane and water at metal surfaces. The key point is that methanol is the simplest organic molecule that contains three types of chemical bonds to be broken, which is an ideal model for studying mode-specific and bond-selective chemistry. As recognized by the other two anonymous reviewers, this is the most complicated gas-surface reaction studied in this detail to date, and qualitatively different from all previously studied systems such as CH_4 and H_2O . This represents a big step beyond CH_4 related reactions, which have been intensively studied and have already deeply advanced our understanding of surface reactions involved polyatomic molecules. Our new findings include the largest vibrational efficacy ever for the C-O stretching mode and the vibrational control of the C-O/C-H branching ratio, which have never been observed before.

Secondly, due to the high permutation symmetry of CH_4 , technically, only a single C-H bond dissociation needs to be considered. For CH_3OH dissociation, however, three competing reaction channels have to be well covered for describing the dynamics, thus spanning a much larger configuration space than that of CH_4 . The development of a globally accurate potential energy surface (PES) for a multi-channel gas-surface reaction is hence not a simple extension of our previous work but indeed poses a big challenge for theorists. For example, we have to use a very large set of permutation invariant polynomials (549 terms) as the input of neural networks, making the training process extremely challenging. Our final PES is of very good quality which accurately describes all reaction channels.

Admittedly, quantum scattering calculations for such a large system is still infeasible. Therefore, quasi-classical trajectory (QCT) calculations were performed at incidence energies above the barrier height for each channel, where the QCT method has proven reliable by previous studies for CHD_3 dissociation on both flat and stepped metal surfaces (see our reply to review #3). On the other hand, the Born-Oppenheimer static-surface approximation definitely affects the absolute values of sticking coefficients, especially at low energies where the barrier change because of surface distortion becomes important, but has a limited impact on the mode-specificity. This can be partially justified according to Refs. 44, 45 and 60.

Taking all together, both fundamentally and technically, we believe that our work goes beyond the extensively studied CH_4 dissociation process and presents the first example of true bond selectivity, to our knowledge, with three different bonds breaking. The new findings will definitely motivate future experimental designs and advance the field. In a broader sense, this work would help to develop more accurate computational methods to understand the selectivity and activity of catalytic reactions, and should be of interest to chemists in a variety of fields.

Reviewer #2 (Remarks to the Author):

The authors have constructed a global full (18) degree-of-freedom potential energy surface for the reaction of CH_3OH on a rigid Cu(111) surface. This is done using a neural network

approach, fitting to over 200,000 DFT energies. These energies are computed using a 3x3 4-layer supercell and a 5x5x1 grid of k-points, which is very reasonable and much more accurate than is typically done for these sorts of big calculations. QCT methods are used to compute the probabilities for C-H, C-O and O-H bond scission as a function of collision energy for a variety of initial vibrational states.

Overall, this is an impressive calculation and the results are very interesting. I recommend publication, for several reasons, after the authors have considered my minor comment below. The paper is well written and should be of interest to chemists in a variety of fields. This is the most complicated gas-surface reaction studied in this manner to date, and a big step beyond CH₄ reactions, which have been intensively studied for almost two decades now. The dissociation of methanol on a metal-based catalyst is an important step in several important processes, and this is the first theoretical study of the dynamics of this reaction. In addition, there has been much interest in bond-selective chemistry, but until now, theory and experiment have focused only on isotopologues of water and methane, such as HOD and CHD₃. That is, they have examined X-H vs X-D bond cleavage. This is the first study, theoretical or experimental, to go beyond this, to my knowledge, with three different bonds breaking. And finally, the vibrational efficacies for promoting dissociation of the various bonds are unusually large. I'm not aware of efficacies this large having been either computed or measured for any system. In addition, the A' and A'' C-H stretching modes have a similar efficacy, which is very unusual. I would have expected the A'' mode to have a very small efficacy. This paper will certainly stimulate much work in the field.

Author reply: We really appreciate the very positive comments from this referee.

1. At two points in the manuscript the authors note that lattice motion effects are likely to modify the dissociation probabilities, listing reference 16 or 14. I think it fair to reference either the original Tiwari/Jackson paper(s), or perhaps the more recent 2016 Guo/Afarjamnia/Jackson Perspective in JPC Letters discussing this topic.

Author reply: This point is well taken. We have replaced the references with the original papers of Jackson and coworkers (Refs. 45 and 46 in the revised manuscript).

Reviewer #3 (Remarks to the Author):

The authors describe a computational study of methanol decomposition dynamics on a Cu(111) surface. They first perform a density functional theory-based electronic structure calculation and fit the resulting grid of calculated points to obtain a high-dimensional potential energy surface for describing the reaction. Reaction dynamics on this potential point to very late barriers relative to the C-H, O-H, and C-O stretching vibrations. Since reactant distortion along each of these coordinates leads to chemically distinct reaction products, the authors conclude that there is a strong likelihood that vibrational state selected beam-surface scattering experiments will uncover unusually strong mode- and bond-selective chemistry. The work is novel in that it advances computational state-of-the-art in molecule-surface

scattering calculations to a molecule that is larger and more structurally complex than any previously studied. Ambitious studies such as this are the drivers that will move the field forward. The work provides a set of detailed predictions that experiment can test. More broadly, the work is at the forefront of attempts to develop increasingly accurate computational methods for predicting barrier heights and reaction paths for important heterogeneously catalyzed processes, and for using those calculations to screen for and predict new catalysts with optimized selectivity and activity. The work also represents an important advance in the ability of computational chemistry to extend chemically accurate predictions of reactivity to larger, and more complex chemical systems.

To enable this calculation, the authors relied on a number of established computational methods to obtain the potential energy of the system in 18D at a number of fixed points, and a recently reported neural-network method to fit a potential to these points. Quasi-classical trajectories are then calculated to gain insight into reaction dynamics. Citations to the relevant precedents are included.

To me, two aspects of the paper really stand out. First, the authors demonstrate that with currently available computational hardware and algorithms, highly detailed and high-dimensional studies of gas-surface reaction dynamics have become accessible to study. Second, I felt that the authors did an excellent job of surveying prior work in the field, and of designing and presenting their work in a way that maximizes its impact on the field.

The authors are to be commended for the extensive scope of their calculation. A strong propensity for mode- and bond-selective chemistry could have been predicted from a much more straightforward transition state calculation and application of Polanyi's rules. Manifestation of that propensity into non-statistical behavior depends on how closely the gas-phase molecule's vibrational eigenstate, which may result from significant state-mixing among zero-order vibrational normal modes, resembles the vibrational state used in the calculation. Non-statistical behavior also depends on the extent of vibrational state mixing that occurs in the entrance channel for the reaction - behavior that arises from, and must be modeled by quantum effects.

Therefore, the quantitative accuracy of the work is likely limited by the same factors associated with prior studies using this general approach. While QCT-based calculations have proven useful for predicting reaction probabilities for trajectories well above reaction barriers, they fail to predict, with quantitative accuracy, reactivity for system energies most important to thermal processes - i.e. trajectories very near in energy to the reaction barrier. Details of vibrational state coupling and energy flow in the entrance channel, which depend strongly on quantum effects missing in the QCT approach, and have been shown to result in decreased accuracy of the computed results. Two recent papers discuss this point, and should be included in the authors' discussion of their results. (D. Migliorini, H. Chadwick, F. Nattino, A. Gutiérrez-González, E. Dombrowski, E. A. High, H. Guo, A. L. Utz, B. Jackson, R. D. Beck, and G.-J. Kroes, *Surface Reaction Barriometry: Methane Dissociation on Flat and Stepped Transition Metal Surfaces* *J. Phys. Chem. Lett.* 8, 4177

(2017 and F. Nattino, D. Migliorini, G.-J. Kroes, E. Dombrowski, E. A. High, D. R. Killelea, and A. L. Utz, Chemically accurate simulation of a polyatomic molecule-metal surface reaction J. Phys. Chem. Lett. 7, 2402 (2016))

In summary, the authors describe an ambitious computational study based on the implementation of established electronic structure and QCT methods. While its significance would have been bolstered by associated experimental work, its detailed predictions and conclusions are exactly what is needed to motivate experimental tests and advance the field.

Author reply: We appreciate the comprehensive and positive comments of this anonymous referee. We agree that the quantitative accuracy of the QCT method is limited at incidence energies near or lower than the corresponding barrier heights. That is the reason that we mainly discussed QCT reaction probabilities at incidence energies above the barrier in this work. As a result, the most important conclusions, *e.g.* the unprecedentedly large efficacy of the C-O stretching mode and the change of branching ratio by vibrational excitation, should be valid in the energy ranges studied here.

According to the referee's suggestions, we reference the two papers to discuss the validation and limitation of the QCT method. Please see page 12 in the revised manuscript, "*As discussed in Refs 12 and 52 on CHD₃ dissociation, the reliability of QCT results decreases when there exist strong vibrational state coupling and energy flow in the entrance channel.*" and page 15, "*In addition, recent QCT based ab initio molecular dynamics applications have reproduced measured dissociative sticking coefficients well above reaction barriers for on CHD₃ on both flat (Ni(111) and Pt(111)) and stepped (Pt(211)) surfaces within chemical accuracy.^{12,52}*"

We also add a statement on page 13 "*We hope these detailed predictions would motivate further experimental tests.*"